# The unique allosteric property of crocodilian haemoglobin elucidated by cryo-EM

Katsuya Takahashi [1], Yongchan Lee [1], Angela Fago [2], Naim M. Bautista[3], Jay F. Storz [3], Akihiro Kawamoto [4], Genji Kurisu [4], Tomohiro Nishizawa [1] ✉ & Jeremy R. H. Tame [1] ✉

The principal effect controlling the oxygen affinity of vertebrate haemoglobins (Hbs) is the allosteric switch between R and T forms with relatively high and low oxygen affinity respectively. Uniquely among jawed vertebrates, crocodilians possess Hb that shows a profound drop in oxygen affinity in the presence of bicarbonate ions. This allows them to stay underwater for extended periods by consuming almost all the oxygen present in the blood-stream, as metabolism releases carbon dioxide, whose conversion to bicarbonate and hydrogen ions is catalysed by carbonic anhydrase. Despite the apparent universal utility of bicarbonate as an allosteric regulator of Hb, this property evolved only in crocodilians. We report here the molecular structures of both human and a crocodilian Hb in the deoxy and liganded states, solved by cryo-electron microscopy. We reveal the precise interactions between two bicarbonate ions and the crocodilian protein at symmetry-related sites found only in the T state. No other known effector of vertebrate Hbs binds anywhere near these sites.

For over 400 million years, haemoglobins (Hbs) from all jawed vertebrates have shared the same tetrameric organisation, with two α and two β subunits[1,2]. These proteins generally fall into two classes, either with intrinsically low oxygen affinity when purified, or with high affinity that is reduced by binding to organic phosphates[3]. The human red blood cell, for example, contains 2,3-diphosphoglycerate (DPG). These heterotropic effectors are non-competitive inhibitors of oxygen binding, which the MWC allosteric model[4] explains by their preferential binding to a low oxygen affinity T state, more constrained by inter-subunit bonds than the high affinity R state[5]. Crocodilian Hbs, however, are governed unusually strongly by carbon dioxide, and their sensitivity to chloride ions and ATP is highly variable[6–8]. Hb from other vertebrates binds carbon dioxide more strongly on deoxygenation (the Haldane effect), but this effect is about twice as strong in crocodilian Hb as in adult human HbA, and independent of pH[9]. In the deoxy state, crocodilian Hbs bind two equivalents of bicarbonate ions per tetramer, with a dissociation constant close to 2 mM, but the oxygenated form of the protein shows no significant bicarbonate binding[2]. While earlier studies did not distinguish between carbon dioxide or bicarbonate ion binding, recent evidence indicates that the same amino group of crocodilian Hbs binds either carbon dioxide covalently (if unprotonated, to form a carbamino group) or bicarbonate noncovalently, if protonated[10–12]. Bicarbonate ions, the predominant form of carbon dioxide in aqueous solution under physiological pH conditions, therefore strongly antagonise oxygen binding, increasing the oxygen tension at half-saturation (p50) of the blood by 40 mmHg and reducing the Hill cooperativity coefficient[2,11].

Crocodilians are descendants of a terrestrial ancestral archosaur, whereas living species such as the American alligator are ambush predators that can hide underwater for extended periods, and which kill their prey by drowning. The adaptations that led to this semiaquatic lifestyle are still being revealed by fossils from the Upper

[1]Graduate School of Medical Life Science, Yokohama City University, Suehiro 1-7-29, Yokohama 230-0045, Japan. [2]Department of Biology, Aarhus University, C. F. Møllers Alle 3, Aarhus DK-8000 Aarhus C, Denmark. [3]School of Biological Sciences, University of Nebraska, 1104 T St., Lincoln NE 68588-0118 NE, USA. [4]Institute for Protein Research, Osaka University, 3-2, Yamadaoka, Suita, Osaka 565-0871, Japan. ✉e-mail: t-2438@yokohama-cu.ac.jp; jtame@yokohama-cu.ac.jp

Jurassic, when the modern crocodilian body plan appeared[13], but the ability to use carbon dioxide to drive almost all the oxygen out of the bloodstream is clearly central to it.

In 1981, Perutz and colleagues proposed that crocodilian Hb had evolved bicarbonate binding by a very limited number of mutations relative to the DPG-binding site of human Hb, at the central cavity of the tetramer, between the N-termini of the β subunits[14]. This speculative model involved bicarbonate ions donating a hydrogen bond to Glu 144 of crocodilian Hb, which replaces Lys 144 of human HbA. Attempts to recreate this binding site by mutating HbA failed to produce any response to bicarbonate, and it was suggested instead by Tame that an entirely novel binding site involved a triplet of positively-charged residues (β38–40) conserved in the C helix of crocodilian β globin[15]. Very unusually, crocodile Hbs also carry a tyrosine residue instead of phenylalanine at position β41 (Fig. S1). Confirmation of this idea was achieved in 1995 by replacing five β subunit residues of human HbA, but a bicarbonate effect as strong as that found in natural proteins was only produced after a further seven mutations were introduced into the α subunits as well[16]. This chimeric protein, called Hb Scuba, shows very high intrinsic oxygen affinity, roughly 10 times that of normal human HbA, and much lower stability. A residue-by-residue approach to test the evolution of bicarbonate binding found that adding the Hb Scuba mutations to a reconstruction of Hb from the ancestral archosaur did not increase bicarbonate sensitivity[12]. Adding ten further residues common to ancestral crocodilian and human HbA gave a protein with 59% of the full bicarbonate effect of ancestral and modern crocodilians, and 21 residues were identified that together produced the full effect[12].

Long-standing efforts to produce well-diffracting crystals of Hb Scuba have failed. Native crocodilian Hbs have also proved refractory to crystallisation[17], but improvements in cryo-electron microscopy (cryo-EM) hardware and software now make it possible, although highly challenging, to solve the structures of proteins as small as Hb, which has a molecular weight of around 64 kDa. Models of oxidised (met) Hb have been determined by cryo-EM using either a Volta phase plate[18], or ultraflat graphene-supported grids[19]. The final resolution of these maps was 3.2 Å and 3.5 Å respectively. 2.8 Å resolution was achieved with more conventional methods by Herzik and colleagues[20]. All of these studies used commercially available lyophilised human Hb, which may have been a factor in limiting resolution. Instead, here we use conventional cryo-EM single particle analysis and fresh protein samples, stored in the liquid state under carbon monoxide to prevent oxidation.

## Results

Alligator Hb (hereafter HbAM) was purified from blood taken from an American alligator (*Alligator mississipiensis*), and stored at 4 °C under carbon monoxide. The preparation of EM grids proceeded smoothly under air with the liganded protein samples, but an anaerobic chamber was required to prepare grids successfully with the deoxy protein in the presence of bicarbonate. For HbAM, the resolution limits of the carbonmonoxy, oxy, and deoxy EM maps are 2.29 Å, 2.31 Å, and 2.20 Å, respectively (Fig. 1a–c and Table 1). Equivalent datasets for human HbA achieved resolution limits of 2.27 Å, 2.24 Å, and 2.35 Å. Diatomic haem ligands were clearly visible in the maps of both HbAM and human HbA (Fig. 1d–g and Table S1), and were modelled with suitable distance restraints[21]. A water molecule is found within the haem pockets of the two deoxy models (Fig. 1h, i).

Like human HbA, HbAM has 141 residues forming seven helices in the α subunits and 146 residues forming eight helices in the β subunits (Fig. S1). The two proteins show 68% sequence identity in the α subunits, and 58% in the β subunits. The quaternary structure is strongly conserved between the two proteins, as shown by the principal allosteric contacts listed in Table S2. Helical regions are traditionally labelled from A to H[22]; the loss of the D helix by α globin is very

ancient[23]. Analyses of animal Hbs demonstrated decades ago that the $\alpha_1\beta_1$ and $\alpha_2\beta_2$ interfaces are relatively fixed on allosteric transition of the protein, but the interfaces between these dimers show significant differences, especially at the so-called "switch" interface between the α subunit C helix and the β subunit FG corner[22,24–26].

The cryo-EM model of deoxy HbAM shows an rmsd of 0.99 Å over all 574 Cα atoms with the crystal structure of deoxy human HbA (PDB 2dn2)[27], and an rmsd of 0.42 Å with our cryo-EM deoxy human HbA model. Overlaying X-ray derived models of liganded human HbA, some of which have missing N-terminal residues, on carbonmonoxy HbAM shows a closer similarity of the R state model PDB 2dn3 (rmsd 1.15 Å over 570 Cα atoms) than the R2 model PDB 1bbb[28] (rmsd 1.71 Å over 570 Cα atoms). The cryo-EM models of oxy and carbonmonoxy human Hb show a rmsd of 0.21 Å with each other over their 566 Cα atoms. Unlike HbAM, the cryo-EM models of liganded human HbA are structurally closer to the R2 form than the R structure, with rmsds of 0.58–0.60 Å and 1.44–1.46 Å, respectively. Comparison of the liganded and unliganded forms of Hb shows a concerted rotation of one αβ dimer relative to the other (Fig. 2). In the case of HbAM, the rotation is 16°, with a translation of 2.4 Å. Crystal structures of R state human HbA show a similar rotation, but slightly smaller translation of 1.3 Å, compared to deoxy HbA[27]. Salt bridges found in T-state human HbA are conserved in HbAM, including the bond formed within each β subunit by Asp 94 (in HbA) or Glu 94 (in HbAM) and His 146, consistent with both proteins having a similar chloride-independent Bohr effect (Fig. S2).

The bicarbonate binding site is deeply buried at the "flexible joint" region of the $\alpha_1\beta_2$ interface formed by the α FG corner and the β C helix (Fig. 3), a location not associated with other heterotropic effectors, which is probably why Perutz discounted it. The hydrogen bonds formed by the ligand are shown in Fig. S3. Almost all of these direct contacts are formed with a single β subunit, including the unique crocodilian residues Lys β38(C4) and Tyr β41(C7). $CO_2$, as well as bicarbonate ions, can bind to the same site of crocodilian Hbs[13], and the side-chain of Lys β38 is clearly well-placed to bind either ligand, by covalent or hydrogen bond, respectively. The α subunit accepts a single hydrogen bond through the carbonyl oxygen of Arg 92, requiring the ligand to donate a hydrogen bond and accounting for the bicarbonate specificity. Introduction of the bulky lysine side-chain in place of Thr β38 displaces the β C helix, including the key allosteric residues Trp β37(C3), and Asn β102 (Fig. 3f). Lys β38 and the conserved Arg β40 sandwich the bicarbonate ion (Fig. 3g, h), suggesting side-chain repulsion in the absence of bicarbonate ions, but Arg β40 is relatively flexible. A number of novel interactions are seen in the deoxy HbAM model however which appear to counter the destabilisation caused by the Lys-Arg-Arg triplet. Tyr β41 forms a hydrogen bond with the main-chain nitrogen of Asp β99. Arg β39 replaces glutamate in human HbA, and points in a direction away from the bicarbonate, to form hydrogen bonds with the carbonyl oxygen atoms of four different residues, Leu β32, Pro β36, Met β48, and Cys β49 (Fig. 3i). Two valine residues of human HbA in this region (β33 and β54) are replaced with isoleucine, while Pro 51 of human HbA is replaced with alanine in HbAM. There is no obvious role for the Gly β29→Ser mutation in Hb Scuba, but the Leu β31→Met mutation is probably required to hold the Lys β38 side-chain in place.

The seven α subunit mutations of Hb Scuba cluster around the CE corner, with Tyr α36 of HbAM (replacing Phe α36 of human HbA) in contact with both Leu α100 and His α103 of the same subunit (Fig. 4a). Komiyama et al. found that without the Leu α100→Phe mutation, Hb Scuba lost not only the bicarbonate effect but all cooperative oxygen binding, an effect that remained entirely unexplained[16]. In the deoxy HbAM model it can be seen that the side-chains of Tyr α36 and Phe α100 lie parallel, but in human HbA Leu α100 pushes Phe α36 slightly towards the $\beta_1$ subunit. This movement would disrupt hydrogen bonds made by Gln β131, as would a side-chain larger

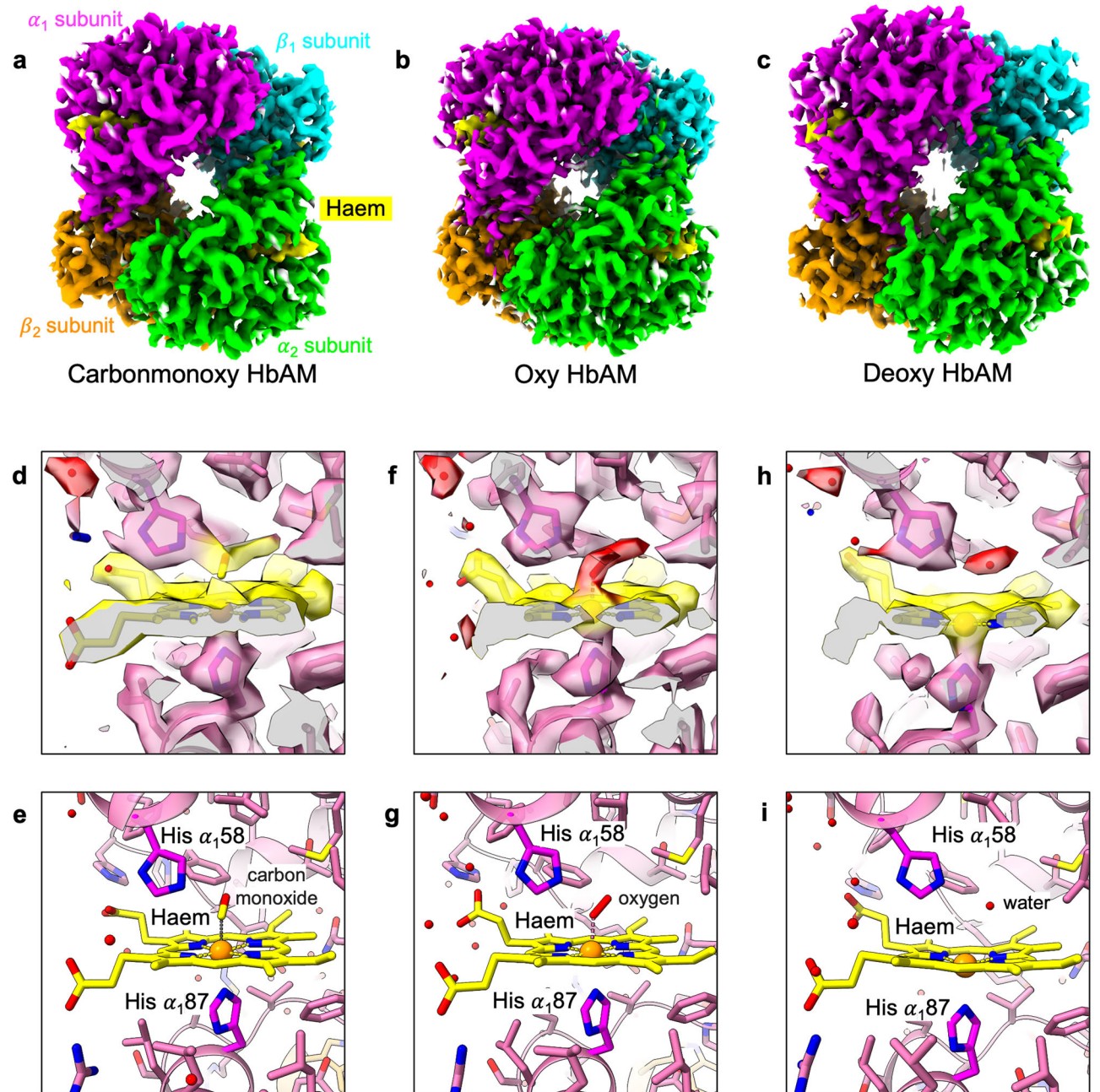

**Fig. 1 | Cryo-EM density maps. a** carbonmonoxy, **b** oxy, and **c** deoxy HbAM, shown with different colours for each subunit. α₁: magenta, α₂: lime, β₁: cyan, β₂: orange, haem: yellow. Cryo-EM density around the haem in the α₁ subunit of carbonmonoxy HbAM (**d**, **e**), oxy-HbAM (**f**, **g**), and deoxy HbAM (**h**, **i**). The carbon atoms of the haem and ligands are shown in yellow. The characteristic doming of the deoxy haem is clearly visible in **h**.

than alanine at α35 (Fig. 4). In the deoxy HbAM structure, Arg β₁135 hydrogen bonds to the carbonyl oxygen of Ala α₁35, another interaction that would be prevented by the presence of Leu α100 pressing against Tyr α36. Even though Arg β135 is only found in Hbs of crocodiles and birds (alanine is generally conserved at this position) it remained entirely overlooked until modelling studies in 2020[8]. The missing Ala→Arg mutation at β135 appears to be at least partly responsible for the instability and high oxygen affinity of Hb Scuba, although allosteric switching to the carbonmonoxy form brings the C-terminal carboxyl group of His β₂146 within 5 Å of the Arg β135 guanidino group, weakening its interaction with Ala α35 (Fig. 4c). Crocodilian Hbs strengthen the α₁β₁ interface by replacing Ala α120 with glutamate, forming additional van der Waals contacts with Ala β₁51.

The carbonmonoxy form of HbAM shows many of the expected features, with the conserved residues of the allosteric core forming the same interactions as in human HbA (Table S2). In switching to the R state, relative motions of the two αβ dimers place the carboxyl group of Asp α₁94 close enough to form hydrogen bonds with the side-chains of Tyr β₂41 and Asn β₂102, roughly where bicarbonate sits in the deoxy T state (Fig. 3j). This tyrosine residue therefore provides an extra hydrogen bond to stabilise the R state. In all other vertebrate Hbs the β subunit C7 residue is phenylalanine, probably due to its role in holding the haem in place[29,30]. α globin has tyrosine at the equivalent position (Tyr α42), which forms an important T state hydrogen bond with Asp β₂99 (Table S2B)[31]. Tyrosine residues in close proximity to haem groups are liable to single electron reduction in the presence of peroxides, giving long-lived tyrosine radicals that can cause oxidative

**Table 1 | Data collection and refinement statistics for all models**

| | CO-HbAM | O₂-HbAM | Deoxy HbAM | CO-HbA | O₂-HbA | Deoxy HbA |
|---|---|---|---|---|---|---|
| **PDB** | **8WIX** | **8WIY** | **8WIZ** | **8WJ0** | **8WJ1** | **8WJ2** |
| **EMDB** | **37571** | **37572** | **37573** | **37574** | **37575** | **37576** |
| **Data collection/processing** | | | | | | |
| Magnification | 165,000 | 165,000 | 165,000 | 165,000 | 165,000 | 165,000 |
| Voltage (kV) | 300 | 300 | 300 | 300 | 300 | 300 |
| Electron exposure (e⁻/Å²) | 60.0 | 60.0/58.4 | 58.4 | 58.4 | 58.4 | 58.4 |
| Defocus range (μm) | −0.4 to −1.2 | −0.4 to −1.2 | −0.4 to −1.2 | −0.4 to −1.2 | −0.4 to −1.2 | −0.4 to −1.2 |
| No. of frames | 54 | 54 | 54 | 55 | 55 | 54/56 |
| Pixel size (Å) | 0.51 | 0.51 | 0.51 | 0.51 | 0.51 | 0.51 |
| Micrographs | 5775 | 16,056 | 12,001 | 8994 | 9002 | 16,579 |
| Initial particles | 7,296,950 | 10,003,848 | 6,675,397 | 4,939,556 | 4,542,816 | 8,682,020 |
| Final particles | 751,964 | 372,582 | 316,544 | 504,650 | 550,843 | 460,786 |
| Symmetry imposed | C2 | C2 | C2 | C2 | C2 | C2 |
| Map resolution (Å) | 2.29 | 2.31 | 2.20 | 2.24 | 2.27 | 2.35 |
| FSC threshold | 0.143 | 0.143 | 0.143 | 0.143 | 0.143 | 0.143 |
| Map sharpening B factor (Å²) | −85.5 | −78.1 | −85.7 | −82.7 | −88.2 | −70.3 |
| **Refinement** | | | | | | |
| Initial model used (PDB code) | - | - | - | 2dn1 | 2dn1 | 2dn2 |
| Model resolution (Å) | 2.3 | 2.3 | 2.2 | 2.2 | 2.3 | 2.3 |
| FSC threshold | 0.5 | 0.5 | 0.5 | 0.5 | 0.5 | 0.5 |
| **Model composition** | | | | | | |
| Non-hydrogen atoms | 4980 | 5162 | 5102 | 4790 | 4732 | 4948 |
| Protein residues | 574 | 574 | 574 | 566 | 566 | 574 |
| Ligands | 8 | 8 | 6 | 8 | 8 | 4 |
| **B factors (Å²)** | | | | | | |
| Protein | 59.2 | 48.0 | 47.8 | 55.3 | 59.3 | 52.9 |
| Ligand | 47.5 | 34.1 | 35.2 | 41.9 | 46.0 | 39.0 |
| **R.m.s. deviations** | | | | | | |
| Bond lengths (Å) | 0.0142 | 0.0131 | 0.0143 | 0.0140 | 0.0143 | 0.0148 |
| Bond angles (°) | 1.80 | 1.79 | 1.87 | 1.84 | 1.85 | 2.03 |
| **Validation** | | | | | | |
| MolProbity score | 2.26 | 2.31 | 2.48 | 2.33 | 2.33 | 2.25 |
| Clashscore | 5.87 | 7.26 | 9.60 | 8.18 | 7.97 | 6.62 |
| Rotamer outliers (%) | 9.58 | 11.67 | 12.08 | 14.54 | 14.98 | 10.82 |
| **Ramachandran plot** | | | | | | |
| Favored (%) | 96.82 | 97.53 | 97.17 | 98.57 | 98.57 | 97.53 |
| Allowed (%) | 2.83 | 2.47 | 2.83 | 1.43 | 1.08 | 2.12 |
| Disallowed (%) | 0.35 | 0.00 | 0.00 | 0.00 | 0.36 | 0.35 |

damage. On the other hand, mutating Phe β41 of HbA to tyrosine provides an extra route for electrons from reductants such as ascorbate to oxidised haem, and such mutations have been explored as means to reduce the toxicity of artificial Hb-based oxygen carriers[32]. Natural selection has apparently firmly chosen Phe C7 for β globin to reduce oxidative damage while demanding Tyr C7 for α globin to stabilize the T state. All diving species face heightened challenges of oxidative damage[33], which may explain the relatively high prevalence of cysteine and histidine residues in crocodilian Hbs. Four histidine residues are common to crocodilian Hbs that are not found in human HbA (Fig. S1); all of them are found at the protein surface, but away from inter-subunit contacts. His α67 and His α113 replace threonine and leucine, respectively, in human HbA. In crocodilian Hbs, residue α68 is either aspartic or glutamic acid, but the structural models offer no convincing evidence of any functional interaction with His α67. His α113 (also found in some bird Hbs) forms a hydrogen bond with Tyr α24 of the same subunit in both deoxy and liganded models, which presumably stabilises the protein but provides little Bohr effect. His β6

and His β56 are uniquely crocodilian residues, replacing glutamate and glycine residues, respectively, not only in human HbA but also in bird Hbs. Both of these histidines are solvent-exposed, providing extra buffering capacity but no additional intramolecular interactions. (β6 is the site of the Glu→Val mutation of sickle-cell Hb.) Individual species may have histidine residues not found generally among crocodilians, for example, His β52 of HbAM or His β136 of the caiman, which are also solvent-exposed, especially His β52, which sits near His β56. It is interesting to speculate that Tyr β41 first arose to accelerate the reduction of oxidized β subunits and was then co-opted as a binding site for bicarbonate ions. Human HbA carrying the mutation Phe β41→Tyr is stable but shows low oxygen affinity due to a more stable T state, possibly due to a hydrogen bond between Tyr β₁41 and the carbonyl of His β₁97[34].

## Discussion
The solution of alligator Hb using cryo-electron microscopy solves several issues regarding Hb structure and evolution. Firstly the models

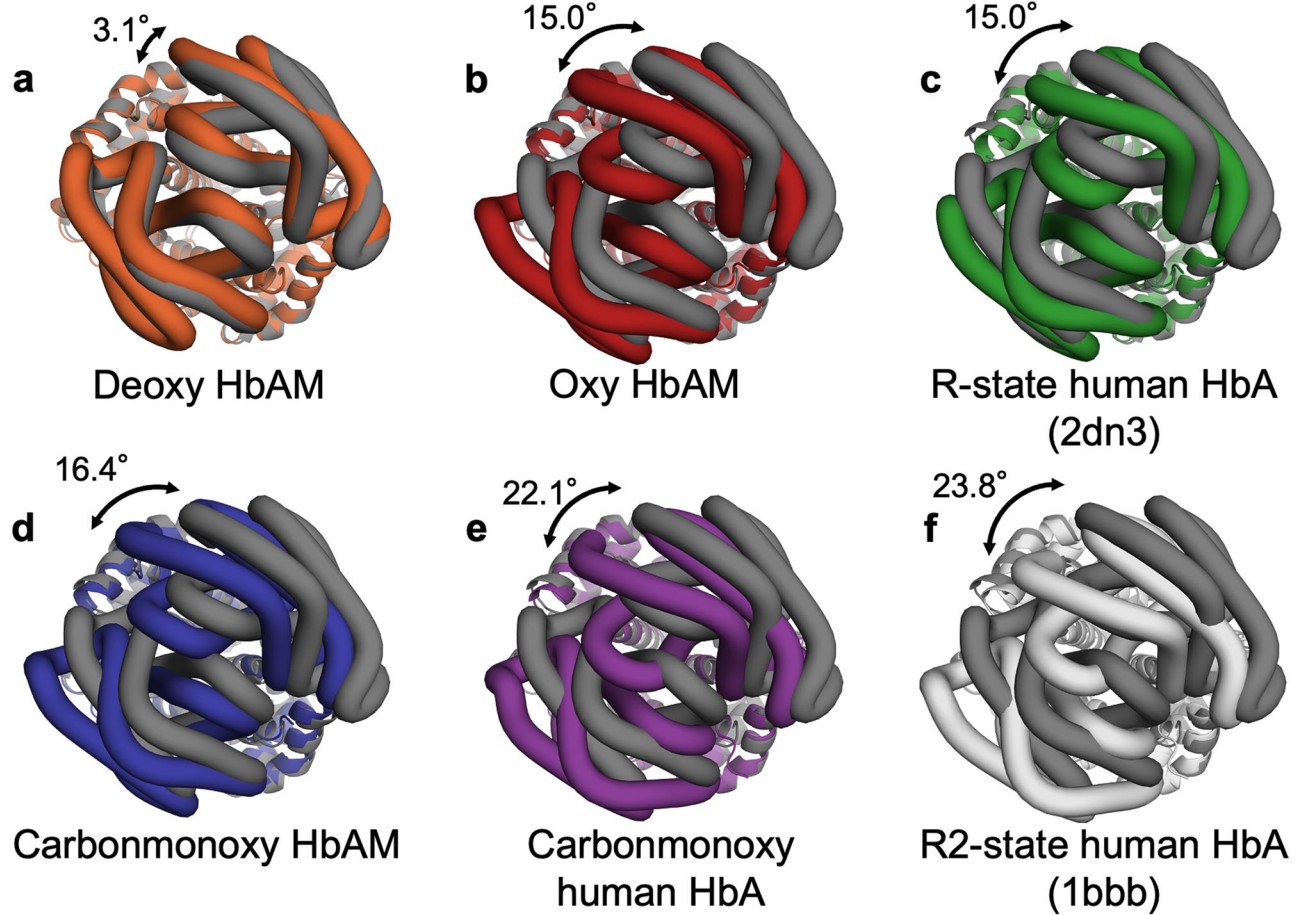

**Fig. 2 | Dimer-dimer rotation angles.** The $\alpha_1\beta_1$ dimers of different Hb models (shown as ribbons) were first superposed on those of the crystallographic model of deoxy human HbA (PDB 2dn2), and then the $\alpha_2\beta_2$ dimers (shown as tubes) were overlaid. The rotation angle of the second least-squares fit is indicated. The deoxy human HbA model is shown in dark grey. **a** deoxy HbAM, orange. **b** oxy-HbAM, red. **c** crystal model of R-state carbonmonoxy human HbA (PDB 2dn3), green. **d** carbonmonoxy HbAM, blue. **e** carbonmonoxy human HbA, purple. **f** crystal model of R2-state carbonmonoxy human HbA (PDB 1bbb), white.

corroborate the X-ray crystallographic structures of human HbA, particularly in the liganded form[27]. Considerable debate arose with the suggestion that the R-state structures of liganded Hb were artefacts arising from the high ionic strength used in crystallisation[35]. The models here support the view that the R2 structural models represent a corner of the volume of conformational space occupied by oxy- or carbonmonoxy-Hb, and that the thermodynamic R-state is well represented by crystallographic models such as PDB 2dn1[27,36,37]. Secondly, the structures of alligator Hb reported here, determined by a different analytical technique, underline the conservation of the allosteric mechanism of Hb over hundreds of millions of years of evolution. Some decades ago, Perutz suggested that most of the amino acid differences between vertebrate Hbs are neutral or nearly so, and that adaptive mechanisms have evolved through a few key replacements[38]. This idea certainly fits the very low heat of oxygenation of Hb from woolly mammoths, the first protein resurrected from an extinct animal[39,40]. Mammoths evolved from equatorial ancestors, but quickly adapted a Hb with thermal properties like those of modern polar species through a single principal mutation[41]. Some bird species also provide examples where single $\alpha$ or $\beta$ subunit mutations yield the higher oxygen affinity required for high-altitude flight[42–45]. Perutz was unable though to explain the extreme pH sensitivity of many fish Hbs, which appears to have been acquired and lost many times, by a variety of mutations affecting either the $\alpha$ or $\beta$ subunits[46–49]. Diving animals such as penguins are able to exploit their body oxygen stores very effectively by regulating Hb with organic phosphates and pH[50], but the

bicarbonate effect of crocodilian Hb is completely unique. Although the same mechanism could function perfectly well in other vertebrates, apparently only the lifestyle of crocodiles gave them sufficient benefit to evolve it; not only does the bicarbonate effect permit extended diving[51], but it may play an important role in unloading oxygen from the blood during the post-prandial "alkaline tide" caused by HCl secretion sufficient to acidify the stomach enough to digest bone[52]. Experimental testing of resurrected ancestral Hbs revealed that the gain of bicarbonate sensitivity and the concomitant loss of ATP-sensitivity occurred in the line of descent leading from the ancestor of birds and crocodilians in the mid-Triassic to the common ancestor of modern crocodiles (~80 million years ago)[12]. Ontology may recapitulate this evolutionary process, since the American alligator produces embryonic (Hb I) and adult (Hb II) isoforms with the same $\alpha$ chain, but with $\beta$ subunits that share only 59% sequence identity[53]. Hb I lacks all the $\beta$ subunit residues associated with bicarbonate binding; it has high oxygen affinity and also strong sensitivity to ATP, which may help developing embryos survive hypoxic conditions in the nest[54].

Our HbAM models suggest that the creation of the bicarbonate binding site disturbs key residues responsible for the delicate balance between the R and T states of the protein, as reflected by the high oxygen affinity and instability of Hb Scuba. Only through a number of mutations is this balance restored. Overall the allosteric controls of Hb appear more fragile than its structure. Changing Thr $\beta38\rightarrow$Lys does not prevent cooperative oxygen binding, but apparently conservative mutations such Phe $\alpha100\rightarrow$Leu eliminate the bicarbonate effect.

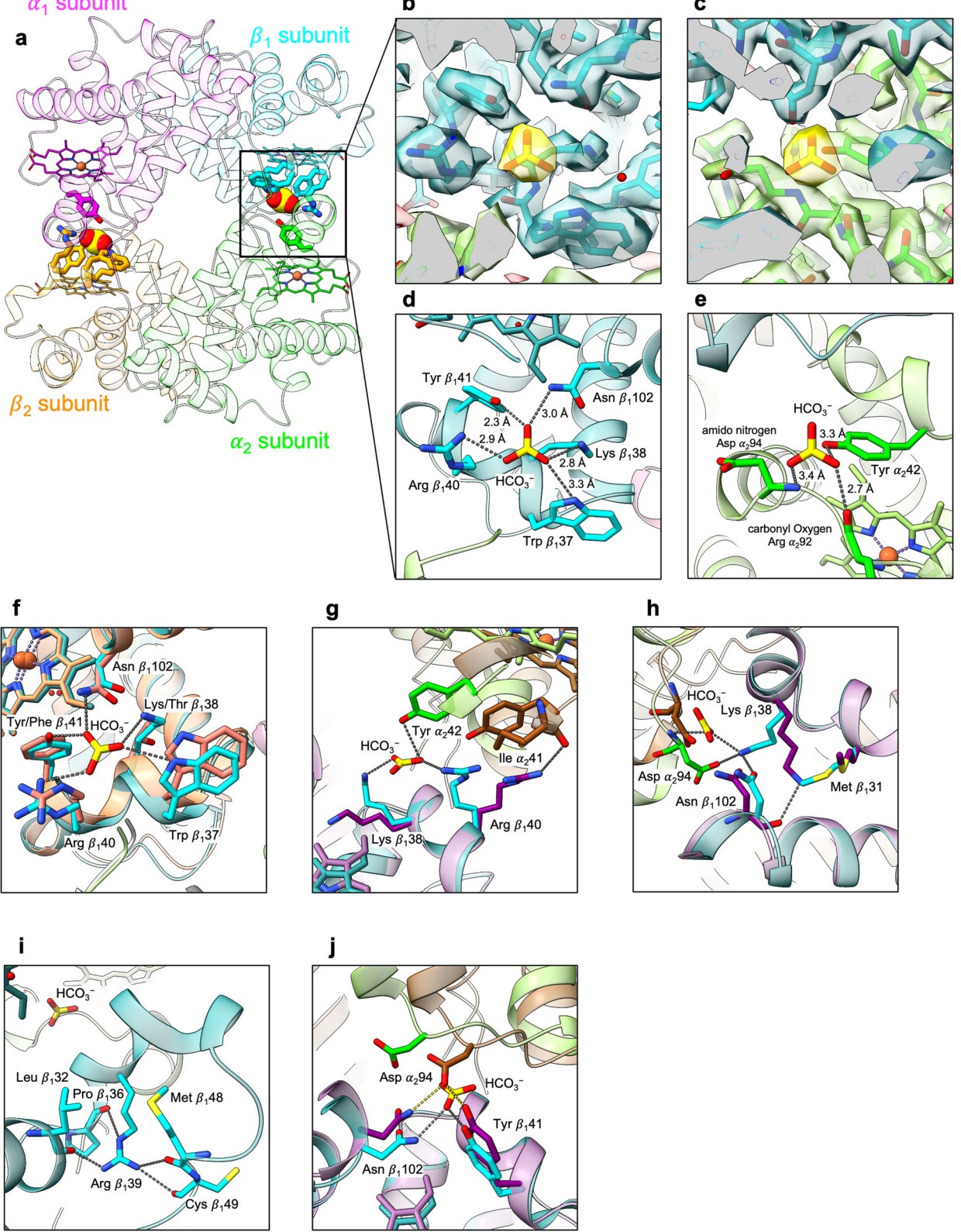

**Fig. 3 | The bicarbonate ion binding site. a** Symmetry-related bicarbonate ions are shown (as CPK models) in the deoxy HbAM tetramer, and side-chains bonding to them are shown as sticks. The carbon atoms of the α₁ subunit are shown in magenta, the β₁ subunit in cyan, the α₂ subunit in lime, and the β₂ subunit in orange. **b**, **c** Cryo-EM density around the bicarbonate ion and (**d**, **e**) the bonding interactions. The hydrogen bonds between the bicarbonate ion and side-chains are shown as grey dotted lines (with distances). **f** An overlay of deoxy HbAM and human HbA at the bicarbonate ion binding site. For deoxy human HbA, the carbon atoms of the β₁ subunit are coloured salmon, and the α₂ subunit is shown in white. **g** α₁β₁ overlay of deoxy HbAM and carbonmonoxy HbAM at the bicarbonate ion binding site, showing relative displacements of Lys β₁38 and Arg β₁40. For carbonmonoxy HbAM, the carbon atoms of the β₁ subunit are shown in purple, and those of the α₂ subunit are shown in brown. **h** Another view of the overlay in **g**, showing Met β₁33. **i** Interactions formed by Arg β₁39, which points away from the bicarbonate ion. **j** A different view of the overlay in **g**, showing Asp α₂94 and Tyr β₁41.

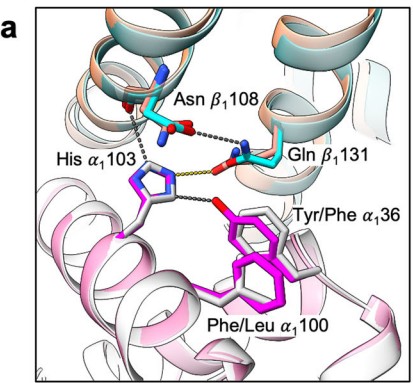
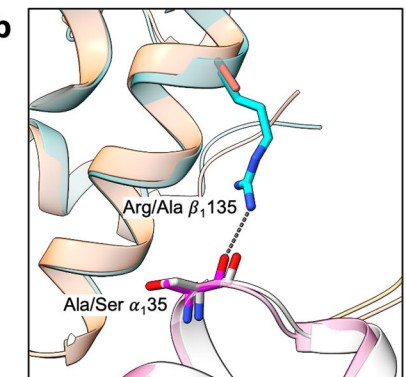
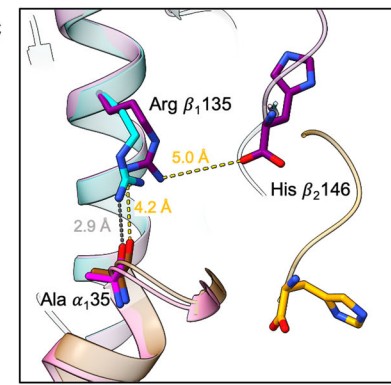

**Fig. 4 | Interactions at the α₁β₁ subunit interface. a** An overlay of deoxy HbAM and human HbA around His α₁103. The carbon atoms of the α₁ and β₁ subunits of HbAM are shown in magenta and cyan, respectively, and those of the α₁ and β₁ subunits of human HbA are shown in white and salmon. **b** the BC corner of the α₁ subunit and the H helix of the β₁ subunit. The hydrogen bond formed by Arg β₁135 and Ala α₁35 in deoxy HbAM is shown. **c** The α₁ BC corner, β₁ H helix, and the C terminus of β₂ subunit of deoxy and carbonmonoxy HbAM. The carbon atoms of the α and β subunits of carbonmonoxy HbAM are shown in brown and purple, respectively. Hydrogen bonds formed by Arg β₁135 are shown as dotted lines, either silver (in deoxy HbAM) or gold (in carbonmonoxy HbAM).

Further work will be required to understand the stability and utility of intermediate Hbs on the evolutionary pathway that led to this unique allosteric control mechanism, but the models described here will hopefully be of immense value in the search for clinically useful Hb-based blood substitutes[55]. The normal levels of bicarbonate ions in both arterial and venous human blood exceed 20 mM[56], more than ten times the bicarbonate binding constant of deoxy crocodilian Hb[2]. Recently there have been a number of clinical trials to evaluate globin gene therapy for patients with thalassemias or sickle-cell disease[57], and two treatments received FDA approval in late 2023 (https://www.fda.gov/news-events/press-announcements/fda-approves-first-gene-therapies-treat-patients-sickle-cell-disease). For patients with some conditions, such as limited lung function, it may be that normal human HbA can be improved upon by borrowing functionality such as that currently unique to crocodilians.

## Methods

### Ethical statement

Human hemoglobin was purified by Prof. Tsuneshige of Hosei University, Japan, using standard protocols and blood freshly drawn from a human volunteer, from whom written informed consent was obtained in accordance with Hosei University Ethical Regulations for Research Involving Human Subjects (Regulation #1141). The methodology was approved by the relevant committee. Whole alligator blood used for this study was obtained by Prof. Dane Crossley of The University of North Texas from nine female adult alligators that were allocated to independent research projects. After euthanasia, a protocol for blood collection was carried out following the Institutional Animal Care Guidelines at The University of North Texas (UNT IACUC #20009). Blood was carried out with the express knowledge and permission of the relevant board at UNT.

**Human HbA sample preparation.** HbA was purified by Prof. Tsuneshige of Hosei University, Japan, using standard protocols and blood freshly drawn from a human volunteer. The washed red cells were bubbled with carbon monoxide before lysis, and the purified protein was shipped to Yokohama at 4 °C under carbon monoxide in stoppered tubes. Carbonmonoxy-HbA was diluted to 5.3 mg/mL with PBS (phosphate-buffered saline) to prepare EM samples. Oxy-HbA was obtained from carbonmonoxy-HbA by passing wet oxygen gas (99.5%) over the protein sample in a round-bottomed flask on ice. The sample was rotated under illumination from a table lamp, until the visible absorbance spectrum indicated oxygenation was complete. Oxy-HbA was finally diluted to 7.0 mg/mL with PBS. 3 μL drops of the liganded

Hb were applied to grids (Quantifoil Cu/Rh R1.2/1.3 carbon holey 300 mesh), which were glow-discharged in advance, and blotted for 3 s with a blot force of 10 at 6 °C and 100% humidity with using Vitrobot (Mark IV, Thermo Fisher), and then was plunge-frozen into liquid ethane.

We failed to prepare suitable deoxy HbA grids under air. In this method, a 3 μL drop of HbA solution was applied to the grid directly after 20 mM fresh sodium dithionite was added to Oxy-HbA (7.0 mg/mL). The grid was blotted and frozen as described above. Despite changes to the absorbance spectrum measured by Nanodrop (Thermo Fisher), which showed an absorbance peak at 550 nm, the initial 3D map (reconstructed from about 30k particles) had the same structure as R-state HbA, suggesting the protein reoxygenated during the vitrification. Deoxy HbA grids were therefore prepared with a Vitrobot installed in an anaerobic chamber, with an oxygen concentration of ~10–20 ppm. Fresh 0.2% (w/v) of dithionite was added to oxy-HbA in the anaerobic chamber, giving a final concentration of 5.5 mg/mL. 2 mM IHP (inositol hexaphosphate) was also added to stabilize the T-state structure. The UV-visible spectrum of the sample was measured before it was applied to the grids and plunge-frozen in liquid ethane. The grids were glow-discharged before being brought into the anaerobic chamber.

**HbAM sample preparation.** Whole blood used for this study was obtained from nine adult alligators that were allocated to research projects in which euthanasia was an endpoint. After the euthanasia protocol was performed following the Institutional Animal Care Guidelines at The University of North Texas (UNT IACUC), 5 mL of blood was collected in heparinized syringes by direct puncture of the ventricle and then immediately stored at −80 °C until further use. Frozen blood was thawed on ice, centrifuged for 5 min, $4000 \times g$ at 4 °C, and supernatant plasma was discarded. The red blood cells were washed 3–5 times with ice-cold 0.9% NaCl containing 0.5 mM EDTA, before being lysed by 1:5 dilution in chloride-free 10 mM HEPES pH 7.6, 0.5 mM EDTA, and incubated on ice for 45 min. The hemolysate was then centrifuged at 4 °C ($12,000 \times g$, 20 min) to remove cell debris. To remove small molecular weight molecules, including allosteric effectors of Hb, the hemolysate was then stripped by passing it through a PD-10 SephadexTM G-25 M desalting column (GE Healthcare) previously equilibrated with 10 mM HEPES pH 7.6, 0.5 mM EDTA.

HbAM was purified by ion exchange chromatography using a Hitrap Q XL column (Cytiva, Uppsala) equilibrated with 20 mM HEPES pH 7.6, 0.5 mM EDTA, and then eluted with a 0–0.1 M NaCl linear gradient, at a flow rate of 0.5 ml/min, at room temperature.

Absorbance was monitored at 415 nm and 280 nm to identify haem-containing proteins. The adult Hb fraction was concentrated by centrifugation ($11,000 \times g$, 10 min), passed again through a PD-10 column, and dialyzed against 10 mM HEPES pH 7.4, 0.5 mM EDTA to remove NaCl. After dialysis, the sample was equilibrated with CO, flash-frozen in liquid nitrogen, and stored at −80 °C until further analysis.

CO-bound HbAM was diluted to a final concentration of 6.5 mg/mL with TN buffer (20 mM Tris and 100 mM NaCl pH 8.0), containing 10 mM of DTT (dithiothreitol). 3 μL of the sample was applied onto the grids (Quantifoil Cu R1.2/1.3 carbon holey 300 mesh), which were glow-discharged at 10 mA for 50 s in advance. The grids were blotted for 3 s with a blot force of 10, and plunge-frozen in liquid ethane with a Vitrobot (Mark IV, Thermo Fisher) at 6 °C and 100% humidity under air. DTT was added to the solution to prevent disulfide bonds and consequent aggregation. Oxy-HbAM was prepared from CO-bound HbAM as described above for human HbA. The UV-Vis spectrum was monitored to confirm that oxygen had completely replaced carbon monoxide. Immediately after sample preparation, oxy-HbAM was diluted to a final concentration of 7.0 mg/mL with DTT-contained TN buffer applied onto the grids and plunge-frozen in the same way as above. Deoxy HbAM EM grid samples were prepared inside an anaerobic chamber, in which the oxygen concentration was kept at 10–20 ppm. Oxy-HbAM was diluted to 5.5 mg/mL with TN buffer containing 10 mM DTT, 10 mM sodium bicarbonate, and 0.2% (w/v) fresh sodium dithionite. Absorbance at 550–570 nm was monitored to confirm the removal of oxygen before a 3 μL drop of protein was applied to the grid and plunge-frozen by Vitrobot inside the anaerobic chamber.

**Data collection.** All cryo-EM data were collected in RIKEN Yokohama, using 300 kV acceleration voltage, with a Titan Krios G4 (Thermo Fisher) equipped with a K3 detector (Gatan) in counting mode with correlative double sampling (CDS), using automated EPU software (ThermoFisher). Images were acquired at a ×160,000 magnification with a pixel size of 0.51 Å. Movies were collected with 54–56 frames using a range of −0.4 to −1.2 μm defocus with two to three shots per hole. The exposure time was 2.0 s, and the exposure dose was 7.6 e/pix/s; the total exposure was 58.4 e/Å². In some datasets, the exposure dose was 7.8 e/pix/s, and the total exposure was 60.0 e/Å². We collected 5775 micrographs for carboxy-HbAM, 9378 micrographs for oxy-HbA, 12,001 micrographs for deoxy HbAM, 8994 micrographs for carboxy HbA, 9002 micrographs for oxy-HbA, and 16,579 micrographs for deoxy HbA.

**Data processing.** All the datasets were processed using CryoSPARC[58], beginning with motion correction and contrast transfer function (CTF) estimation. Particles were picked automatically using Blob Picker job, extracted from the micrographs with a 256-pixel box, and Fourier cropped to 128 pixels. 2D classification was performed, and Hb-shaped 2D class averages were manually selected to build initial volumes. Three classes of ab initio models were generated, and several bad volumes were used in heterogeneous refinement as decoy to draw particles that are not clearly Hb-shaped. For carbonmonoxy-HbA and oxy-HbA, a prototype 3.9 Å resolution map of HbAM, built from about 49k particles, was used as a reference volume. For deoxy HbA, carbonmonoxy HbAM and deoxy HbAM, a high-resolution map of oxy-HbA was used as a reference volume. For oxy-HbAM, a high-resolution intermediate map of carboxy-HbAM was used as a reference volume. For all datasets, several rounds of heterogeneous refinement were performed, and the particles from the best classes were re-extracted with a 512-pixel box size and Fourier cropped to 256 pixels. Several rounds of refinement were then performed with a homogeneous set of particles, and the final maps were obtained from non-uniform refinement with C2 symmetry and optimized defocus and CTF parameters. The final HbAM maps (Figs. S5–7) were obtained with resolution of: carbonmonoxy, 2.29 Å (from 751,964 final particles), oxy, 2.31 Å

(from 372,582 final particles), deoxy, 2.20 Å (from 316,544 final particles). The final human HbA maps (Figs. S8–10) were obtained with resolution of: carbonmonoxy, 2.24 Å (from 504,650 final particles), oxy, 2.27 Å (from 550,843 final particles), deoxy, 2.35 Å (from 460,786 final particles). The overall resolution of the final maps was determined by the gold-standard Fourier shell correlation (FSC).

**Model building.** The HbAM models were built from an AlphaFold2[59] predicted model, and each subunit of the models was fitted to the relevant map using ChimeraX[60]. The oxy and carbonmonoxy human HbA models were built from the R-state crystal structure (PDB 2dn1), and the deoxy human HbA model was built from the T-state crystal structure (PDB 2dn2) using Coot[61]. Model restraints were determined using ISOLDE[6], and refinement was performed with Servalcat[62]. Manual adjustments were carried out with Coot. The data collection, refinement parameters, and model statistics are summarized in Table 1.

**Modelling bicarbonate ion in deoxy HbAM.** Early steps of the refinement of deoxy HbAM were performed with C1 symmetry. After the map resolution improved beyond 3 Å, extra density was observed at two symmetry-related positions at the $\alpha_1\beta_2$ and $\alpha_2\beta_1$ subunit interfaces. A water molecule was placed in the middle of this density at each position, and model refinement was performed with Servalcat[63], which yielded positive Fo-Fc density (as shown in Fig. S11), where a bicarbonate ion was finally modelled. The final cycles of non-uniform refinement were performed with C2 symmetry applied to the model, which improved the resolution.

**Deoxy human HbA 3D classification.** IHP (inositol hexaphosphate) was added to stabilize deoxy human HbA before vitrification under anaerobic conditions. After several rounds of heterogeneous refinement and non-uniform refinement with C2 symmetry, a 2.35 Å resolution deoxy human HbA map was obtained from about 460k particles. A flat and transverse region of density was observed between the N-terminal regions of the β subunits, where IHP is known to bind. Non-uniform refinement was performed with C1 symmetry, which gave a density map suggesting the phosphate groups could interact with positively-charged amino acid side-chains lining the pocket, but the map was too blurry to show this clearly. To obtain a better map of the phosphate binding site, 3D classification was performed to classify 460k particles into eight classes with a mask around the IHP map. Two major classes were obtained: 169k particles of liganded HbA in class 0 and 134k particles of unliganded HbA in class 1. Each map was locally refined with the same mask, to give a map at 2.59 Å resolution of IHP-liganded deoxy HbA and a separate map at 2.66 Å resolution of IHP-free deoxy HbA (Fig. S12a–c). For the liganded model, IHP was fitted to the map using Coot. IHP sits between Val β1, His β2, Lys β82, and His β143 of both β subunits, apparently without a single strongly preferred conformation. Compared to the crystal structure of deoxy human HbA (2dn2), the two N-terminal residues move towards IHP, and the side-chains of Lys β82 move towards the central cavity to avoid steric hindrance with IHP (Fig. S12d). Our results are broadly in line with previous crystallographic models of deoxy Hb bound to 2,3-diphosphoglycerate (Fig. S12e)[64,65].

**Reporting summary**
Further information on research design is available in the Nature Portfolio Reporting Summary linked to this article.

## Data availability
The data that support this study are available from the corresponding authors upon request. The final models have been deposited in the Protein Data Bank (PDB). The cryo-EM density maps, half maps, and masks have been deposited in the Electron Microscopy Data Bank (EMDB), and the cryo-EM raw images have been deposited in the

**Article**

Electron Microscopy Public Image Archive (EMPIAR). Accession codes are as follows: alligator haemoglobin in carbonmonoxy form (8WIX, EMD-37571, EMPIAR-11988, alligator haemoglobin in oxy form (8WIY, EMD-37572, EMPIAR-11989), alligator haemoglobin in deoxy form (8WIZ, EMD-37573, EMPIAR-11990), human haemoglobin in carbonmonoxy form (8WJ0, EMD-37574, EMPIAR-11991), human haemoglobin in oxy form (8WJ1, EMD-37575, EMPIAR-11992), human haemoglobin in deoxy form (8WJ2, EMD-37576, EMPIAR-11993).

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

## Acknowledgements

This paper is dedicated to our friend and mentor the late Kiyoshi Nagai FRS, who touched many lives with his personal warmth and transformative science. We thank Professors A. Tsuneshige and D. Crossley for their kind gifts of purified human HbA and alligator blood, respectively. K.T. acknowledges a JST university fellowship, grant number JPMJFS2140, Sasakawa Scientific Research Grant from The Japan Science Society, and SUNBOR scholarship. J.F.S. acknowledges grants from the NIH (R01HL087216) and NSF (OIA-1736249). Y.L. acknowledges a grant from JSPS (JP24H02264). T.N. acknowledges grants from JSPS (JP20H03216, JP23H02439, and JP24H02264). A.K. and G.K. acknowledge grants from JSPS (JP21H02417 and JP23H04958). A.F. acknowledges a NOVA grant from the Aarhus University Research Foundation (AUFF-E-2023-9-55). This research was partially supported by the Research Support Project for Life Science and Drug Discovery (Basis for Supporting Innovative Drug Discovery and Life Science Research (BINDS)) from AMED under Grant Number JP23ama121001. The cryo-EM experiments were performed at the RIKEN Yokohama cryo-EM facility and in part at the Institute for Protein Research, Osaka University.

## Author contributions

K.T., Y.L., T.N. collected and processed EM data, and refined models. A.F., N.M.B., and J.F.S. purified samples, A.K. and G.K. prepared deoxy samples. K.T., Y.L., T.N., and J.R.H.T. analysed the models. J.R.H.T. wrote the draft manuscript, which was modified with the input of all authors.

## Competing interests

The authors declare no competing interests.
