## [Peer Review File · Nature Communications]

The unique allosteric property of crocodilian haemoglobin elucidated by cryo-EMREVIEWER COMMENTS

Reviewer #1 (Remarks to the Author):

This manuscript helps lay to rest the precise molecular mechanisms underlying the unique ability of extant Crocodylia hemoglobin to allosterically bind bicarbonate ions that have been speculated on for over 40 years. The structural data and interpretations appear to be sound and support physiological data collected from this animal group both in the 1980's and by several authors of the present study. While I do not dispute the structural components of the study, I do have several questions regarding some of the biological interpretations regarding the evolution of this unusual trait.

First, the authors place considerable emphasis on the proposition that the evolved ability to bind bicarbonate ions is central to the underwater breath-hold ambush predation strategy of members of this group, which is suggested to allow them to almost completely utilize their blood oxygen stores (an alternative proposal is that this mechanism would be beneficial for oxygen offloading during digestion). However, penguins and seals (and presumably deep diving whales) are also able to almost completely exploit their blood oxygen stores despite their hemoglobin being primarily regulated by organic phosphates and protons (Bohr effect). While I agree that the simplicity of the crocodylian allosteric effector system has a certain elegance and logic that would be embraced by engineers, I can see no reason why it is either essential for or has any inherent advantage for diving relative to the phosphate/proton mechanism of other secondarily aquatic vertebrates.

Another peculiarity regarding Crocodylia hemoglobin that was not discussed, but that is clearly relevant to their biology is its very high histidine content. These residues presumably buffer the protons liberated together with bicarbonate during the dissociation of carbonic acid inside the red blood cell— or via carbamate formation at Lys β 38 or the α -chain N-termini—during diving (hence maintaining cellular pH), though it remains unclear why the histidine content is markedly higher than in whales and seals which presumably also deal with high proton loads while submerged. I'm curious as to whether or not the generated structural models could provide any insights regarding the role of these additional histidine residues compared to human HbA? I ask as I see no obvious benefit for this trait during the alkaline tide given the robust band 3 anion exchange ability of alligator red blood cells. However, the ability to use bicarbonate to aid oxygen offloading during digestion (when proton and organic phosphate binding to hemoglobin would not be expected to meaningfully increase) does seem to be of distinct advantage. Though again, snakes are also able to navigate a significant alkaline tide in the absence of a bicarbonate binding system.

I particularly found the discussion regarding Tyr β 41 (lines 183-194) to be interesting. Notably, in this regard, Crocodylia diverged from other archosaurs during a period of the earth's history when atmospheric oxygen levels were dropping precipitously (from ~30 to <15%), which may well have also played a key role in the evolution of this residue substitution and hence the unique allosteric control of hemoglobin in this group though is not discussed.

A final comment regards the suggestion that the alligator structure will be of immense value in the search (development?) of Hb-based oxygen carriers. Unless the authors are referring to modified Crocodylia hemoglobins for this purpose (which is not clear), this comment appears to conflict with results from the recent Natarajan study cited in the manuscript which suggests that transferring this trait to human hemoglobin is unlikely to be successful due to its divergent genetic background. Can the authors please clarify their intent here?

Minor comment:

line 50: 'that same'  'that the same'?

Reviewer #2 (Remarks to the Author):

This manuscript represents the culmination of over 40 years of effort to determine structurally how bicarbonate binding to crocodylian Hbs causes O₂ release during submersion to allow these animals to drown their prey. In addition, the 2.5 Å resolution EM structure determinations of both adult human and alligator hemoglobins in their deoxygenated, O₂ bound and CO bound forms are remarkable experimental achievements, and the results verify the generality of the allosteric mechanism for all vertebrate hemoglobins and general aspects of the alkaline Bohr effect. Thus, this work has my wholehearted support, and reviewing the paper was both enjoyable and satisfying.

The paper is a wonderful tribute to the late Professor Kiyoshi Nagai, who was a great scientist, mentor, and friend to many of the authors and his other colleagues around the world. He guided the initial work on trying to determine the bicarbonate-binding site by site-directed mutagenesis of human hemoglobin (i.e. Hb Scuba) with his students at Cambridge University. The current paper also serves to support and explain in more molecular detail the sequence and mutagenesis studies in reference 12, which mapped out the evolution of the bicarbonate-binding site in crocodylians from a common Archosaur ancestor with birds. Now the story is complete in molecular, evolutionary, and physiological detail.

I have only five very minor suggestions and comments for the authors, which they may or may not want to consider. Again, I congratulate the authors on very well written, concise, but scientifically thorough paper with remarkably clear graphics.

1. Page 1, line 24 and other places. – The authors suggest that bicarbonate should have "universal utility" as potential allosteric effector for O₂ off loading by red cell hemoglobin during strenuous muscle activity and CO₂ production. Clearly that does not appear to be the case for most vertebrates. I wonder if the authors have thought about why that may be the case, particularly for mammals and birds, which produce large amounts of CO₂ during strenuous aerobic activity. My view is that the CO₂ needs to be removed rapidly and continuously from the body if strenuous aerobic activity is to be maintained for long periods of time. Storing bicarbonate in the blood bound to hemoglobin would be detrimental allowing a build up of CO₂. In contrast, the alligators sink to the bottom of rivers with their prey, only maintaining contraction of the jaw muscles. Their need for O₂ may be just to maintain neuronal activity and the jaw contraction. However, perhaps my view is naïve.

2. Page 1, line 30 – I might qualify this sentence by writing: "No other known vertebrate Hb effector binds anywhere..." I am not sure about invertebrate Hbs and bicarbonate binding, particularly among the complex worms with giant hemoglobins.

3. Page 2, line 42 – need commas - "...crocodylian Hbs, however, are..."

3. Page 2, line 47 – I think one of the reasons it has been hard to find the bicarbonate binding site is that the affinity is fairly weak, K_d ~ 2 mM. Compare that value to the K_d's for 2,3-BPG binding to deoxyHbA (or ATP binding to bird Hbs). Thus, the bicarbonate levels do have to be very high in the alligator blood.

4. Page 4 and throughout the paper – Using HbAM for adult Alligator hemoglobin and HbA for adult human hemoglobin is bit awkward. I don't think the authors should change the nomenclature, but I might write out "human HbA" in places where it might be confusing. For example, on line 100 I would write: "Equivalent data sets for human HbA achieved..."

5. Page 8, lines 180 and 181. Is there any experimental evidence that beta PheC7 helps to hold heme in place (i.e. direct measurements of the rate of heme loss for beta PheC7 Hb mutants)? If so, a reference should be given.

Reviewer #3 (Remarks to the Author):

Takahashi et al describe the cryo-EM structures of crocodilian haemoglobin as well as human haemoglobin in different states. The authors show for the first time the allosteric binding site of bicarbonate.

The cryo-EM structure work done on this paper looks solid, and even though bicarbonate is a very small ligand, the evidence for the binding site seems convincing.

I have only minor comments:

- Scale bars on the micrographs and 2D class averages shown in the supplementary figures should be included.

- What type of particle picking was used? The authors say they used automatic particle picking, does that mean "blob picker" or template picking or perhaps Topaz. Please specify in the method section.

- It is unclear what volumes were used as input for the heterogeneous refinements. The authors explain the use of one volume, but it looks like on the figures that they used the ab-initio generated models as input as well, but it's unclear and I think this should be further clarified.

Overall, I would like to congratulate the authors on a well-written and interesting paper, and with the minor edits proposed I believe it is ready for publication.

Responses to reviewers.

Numbered responses in black follow Reviewer comments in red.

Reviewer #1 (Remarks to the Author):

This manuscript helps lay to rest the precise molecular mechanisms underlying the unique ability of extant Crocodylia hemoglobin to allosterically bind bicarbonate ions that have been speculated on for over 40 years. The structural data and interpretations appear to be sound and support physiological data collected from this animal group both in the 1980's and by several authors of the present study. While I do not dispute the structural components of the study, I do have several questions regarding some the biological interpretations regarding the evolution of this unusual trait.

First, the authors place considerable emphasis on the proposition that the evolved ability to bind bicarbonate ions is central to the underwater breath-hold ambush predation strategy of members of this group, which is suggested to allow them to almost completely utilize their blood oxygen stores (an alternative proposal is that this mechanism would be beneficial for oxygen offloading during digestion). However, penguins and seals (and presumably deep diving whales) are also able to almost completely exploit their blood oxygen stores despite their hemoglobin being primarily regulated by organic phosphates and protons (Bohr effect). While I agree that the simplicity of the crocodylian allosteric effector system has a certain elegance and logic that would be embraced by engineers, I can see no reason why it is either essential for or has any inherent advantage for diving relative to the phosphate/proton mechanism of other secondarily aquatic vertebrates.

1) The reviewer is correct that other diving animals are able to use their oxygen stores highly efficiently without the bicarbonate effect, and to emphasise this we have added a new sentence and citation (reference 50). We do not suggest the bicarbonate effect of crocodiles is superior to the evolutionary routes taken by penguins and whales, but it does indeed offer a more direct solution to protein engineers wishing to make artificial or modified blood for patients of strongly impaired lung function. Direct evidence has been obtained to show the bicarbonate effect is involved in the diving ability of crocodylians, and the relevant paper is now cited (reference 51).

Another peculiarity regarding Crocodylia hemoglobin that was not discussed, but that is clearly relevant to their biology is its very high histidine content. These residues presumably buffer the protons liberated together with bicarbonate during the dissociation of carbonic acid inside the red blood cell—or via carbamate formation at Lys β 38 or the α -chain N-termini—during diving (hence maintaining cellular pH), though it remains unclear why the histidine content is markedly higher than in whales and seals which presumably also deal with high proton loads while submerged. I'm curious as to whether or not the generated structural models could provide any insights regarding the role of these additional histidine residues compared to human HbA? I ask as I see no obvious benefit for this trait during the alkaline tide given the robust band 3 anion exchange ability of alligator red blood cells. However, the ability to use bicarbonate to aid oxygen offloading during digestion (when proton and organic phosphate binding to hemoglobin would not be expected to meaningfully increase) does seem to be of distinct advantage. Though again, snakes are also able to navigate a significant alkaline tide in the absence of a bicarbonate binding system.

2) The referee is absolutely correct when suggesting that crocodylian Hbs have higher histidine content (as can be checked using Figure S1). We now state that both cysteine and histidine residues are unusually abundant, and provide several sentences regarding histidine residues common to crocodylian Hbs, but absent from human HbA.

I particularly found the discussion regarding Tyr β 41 (lines 183-194) to be interesting. Notably, in this regard, Crocodylia diverged from other archosaurs during a period of the earth's history when atmospheric oxygen levels were dropping precipitously (from ~30 to <15%), which may well have also played a key role in the evolution of this residue substitution and hence the unique allosteric control of hemoglobin in this group though is not discussed.

3) The reviewer mentions changes in global levels of atmospheric oxygen hundreds of millions of years ago, but it is perhaps too speculative to suggest these pushed crocodylians in a unique evolutionary direction.

A final comment regards the suggestion that the alligator structure will be of immense value in the search (development?) of Hb-based oxygen carriers. Unless the authors are referring to modified Crocodylia hemoglobins for this purpose (which is not clear), this comment appears to conflict with results from the recent Natarajan study cited in the manuscript which suggests that transferring this trait to human hemoglobin is unlikely to be successful due to its divergent genetic background. Can the authors please clarify their intent here?

4) Our final comment regarding Hb-based oxygen carriers reflects attempts both by genetic modification to introduce variant globin genes into patients, and producing a bottled form of Hb. The latter has proved immensely difficult so far but is not entirely impossible. Given the problems with both approaches, we have replaced "unquestionably" with "hopefully", which covers the reviewer's concern, and provided additional sentences and citations related to current efforts at gene therapy for Hb related diseases.

Minor comment:

line 50: 'that same'  'that the same'?

5) Corrected

Reviewer #2 (Remarks to the Author):

This manuscript represents the culmination of over 40 years of effort to determine structurally how bicarbonate binding to crocodylian Hbs causes O₂ release during submersion to allow these animals to drown their prey. In addition, the 2.5 Å resolution EM structure determinations of both adult human and alligator hemoglobins in their deoxygenated, O₂ bound and CO bound forms are remarkable experimental achievements, and the results verify the generality of the allosteric mechanism for all vertebrate hemoglobins and general aspects of the alkaline Bohr effect. Thus, this work has my wholehearted support, and reviewing the paper was both enjoyable and satisfying.

The paper is a wonderful tribute to the late Professor Kiyoshi Nagai, who was a great scientist, mentor, and friend to many of the authors and his other colleagues around the world. He guided the initial work on trying to determine the bicarbonate-binding site by site-directed mutagenesis of human hemoglobin (i.e. Hb Scuba) with his students at Cambridge University. The current paper also serves to support and explain in more molecular detail the sequence and mutagenesis studies in reference 12, which mapped out the evolution of the bicarbonate-binding site in crocodylians from a common Archosaur ancestor with birds. Now the story is complete in molecular, evolutionary, and physiological detail.

I have only five very minor suggestions and comments for the authors, which they may or may not want to consider. Again, I congratulate the authors on very well written, concise, but scientifically thorough paper with remarkably clear graphics.

1. Page 1, line 24 and other places. – The authors suggest that bicarbonate should have "universal utility" as potential allosteric effector for O₂ off loading by red cell hemoglobin during strenuous muscle activity and CO₂ production. Clearly that does not appear to be the case for most vertebrates. I wonder if the authors have thought about why that may be the case, particularly for mammals and birds, which produce large amounts of CO₂ during strenuous aerobic activity. My view is that the CO₂ needs to be removed rapidly and continuously from the body if strenuous aerobic activity is to be maintained for long periods of time. Storing bicarbonate in the blood bound to hemoglobin would be detrimental allowing a build up of CO₂. In contrast, the alligators sink to the bottom of rivers with their prey, only maintaining contraction of the jaw muscles. Their need for O₂ may be just to maintain neuronal activity and the jaw contraction. However, perhaps my view is naïve.

1) The reviewer makes an excellent point regarding strenuous aerobic activity, but such questions can only be studied by whole animal biology. The added sentences and citation of bicarbonate levels in human blood (new reference 56) show that the bicarbonate effect may well indeed have universal utility (see point 3 below).

2. Page 1, line 30 – I might qualify this sentence by writing: "No other known vertebrate Hb effector binds anywhere..." I am not sure about invertebrate Hbs and bicarbonate binding, particularly among the complex worms with giant hemoglobins.

Page 2, line 42 – need commas - "...crocodilian Hbs, however, are..."

2) We accept the reviewer's point, and have included the words "of vertebrate Hbs" in the sentence. Commas added as suggested.

3. Page 2, line 47 – I think one of the reasons it has been hard to find the bicarbonate binding site is that the affinity is fairly weak, K_d ~ 2 mM. Compare that value to the K_d's for 2,3-BPG binding to deoxyHbA (or ATP binding to bird Hbs). Thus, the bicarbonate levels do have to be very high in the alligator blood.

3) This is also an important point, but in fact bicarbonate binding affinity of crocodilian Hb is comparable to the DPG binding affinity of human HbA. The difference is that DPG is confined within the red cell. Extra sentences and citations added at the end of the manuscript show that the bicarbonate binding is quite strong enough to be relevant in a human patient.

4. Page 4 and throughout the paper – Using HbAM for adult Alligator hemoglobin and HbA for adult human hemoglobin is bit awkward. I don't think the authors should change the nomenclature, but I might write out "human HbA" in places where it might be confusing. For example, on line 100 I would write: "Equivalent data sets for human HbA achieved..."

4) We accept the reviewer's point and have replaced "HbA" with "human HbA" throughout the paper.

5. Page 8, lines 180 and 181. Is there any experimental evidence that beta PheC7 helps to hold heme in place (i.e. direct measurements of the rate of heme loss for beta PheC7 Hb mutants)? If so, a reference should be given.

5) Two papers from the Olson group on heme retention of globin mutants are now cited.

Reviewer #3 (Remarks to the Author):

Takahashi et al describe the cryo-EM structures of crocodilian haemoglobin as well as human haemoglobin in different states. The authors show for the first time the allosteric binding site of bicarbonate.

The cryo-EM structure work done on this paper looks solid, and even though bicarbonate is a very small ligand, the evidence for the binding site seems convincing.

I have only minor comments:

- Scale bars on the micrographs and 2D class averages shown in the supplementary figures should be included.

1) Scale bars are now included in the amended figures.

- What type of particle picking was used? The authors say they used automatic particle picking, does that mean “blob picker” or template picking or perhaps Topaz. Please specify in the method section.

2) The methods section now clearly explains the picking procedure.

- It is unclear what volumes were used as input for the heterogenous refinements. The authors explain the use of one volume, but it looks like on the figures that they used the ab-initio generated models as input as well, but it's unclear and I think this should be further clarified.

3) Volumes used for refinement are now explained in the revised supplementary information.

Overall, I would like to congratulate the authors on a well-written and interesting paper, and with the minor edits proposed I believe it is ready for publication.

4) We are grateful to all the reviewers for their highly constructive suggestions and support for publication.